# Foliar Symptomology, Nutrient Content, Yield, and Secondary Metabolite Variability of Cannabis Grown Hydroponically with Different Single-Element Nutrient Deficiencies

**DOI:** 10.3390/plants12030422

**Published:** 2023-01-17

**Authors:** David Llewellyn, Scott Golem, A. Maxwell P. Jones, Youbin Zheng

**Affiliations:** 1School of Environmental Science, University of Guelph, Guelph, ON N1G 2W1, Canada; 2HEXO Corp., 120 Chem. de la Rive, Gatineau, QC J8M 1V2, Canada; 3Department of Plant Agriculture, University of Guelph, Guelph, ON N1G 2W1, Canada

**Keywords:** *Cannabis sativa*, cannabinoid, plant images, controlled environment, fertilizer, nutrition, tissue analysis

## Abstract

In controlled environment production systems, *Cannabis sativa* (hereafter cannabis) is a commodity with high nutrient demands due to prolific growth under optimized environmental conditions. Since nutrient deficiencies can reduce yield and quality, cultivators need tools to rapidly detect and evaluate deficiency symptoms so corrective actions can be taken quickly to minimize losses. We grew cannabis plants in solution culture with different individual nutrient elements withheld from the solutions to identify deficiency symptoms. Control plants received a complete nutrient recipe, whereas the following single elements were withheld from the respective nutrient deficiency treatments: N, P, K, Ca, Mg, S, Fe, and Mn. The nutrient treatments began when the photoperiod was switched to a 12/12 h (light/dark), and plants were grown to commercial maturity. Plants were monitored daily, and the development of visual deficiency symptoms were recorded. Photographs of each plant were taken weekly. Upon the onset of visual deficiency symptoms, both upper- and lower-canopy foliage were analyzed for nutrient element concentrations. At harvest, plants were evaluated for biomass partitioning, and the cannabinoid composition of inflorescence tissues. This manuscript describes the onset and progression of nutrient deficiency symptoms (with pictures), relates symptomology to foliar nutrient analyses, and contextualizes the relationships between nutrient deficiencies and cannabis growth, yield, and quality. Aboveground vegetative fresh weights were reduced by 73% in the -N treatment and 59% in the -P treatment, compared with the control. All deficiency treatments except for -Fe and -Mn had floral yields reduced by between 33% to 72%, compared with the control. Overall, deficiencies of individual nutrients can substantially reduce vegetative growth and inflorescence yield, although only minor effects were observed in secondary metabolite composition. The onset of individual deficiency symptoms did not always correspond with elemental analysis of foliar tissues. Cultivators should take an integrated approach in diagnosing nutrient deficiencies and take timely corrective actions to optimize productivity and minimize losses to yield and quality.

## 1. Introduction

In a controlled environment *Cannabis sativa* (hereafter cannabis) production and cultivators can monitor and control a myriad of crop production inputs that will affect cannabis yield and quality (Backer et al., 2019) [1]. In these environments, cannabis has relatively high nutrient demands to support prolific growth of vegetative and reproductive tissues under optimized environmental conditions (e.g., temperature, VPD, light intensity, and CO_2_ levels). While research on cannabis fertility is ongoing, optimum fertilizer levels are highly dependent on the characteristics of individual cultivation systems including specific cultivar demands, planting density, light intensity, CO_2_ concentration, type and size of substrate, and irrigation methods (Resh 2012; Zheng 2022) [2,3]. Nutrient deficiencies commonly develop due to shortages or imbalances in the crop inputs (e.g., makeup of the fertigation solution, growing substrate, fertilizer additives, etc.). However, even when a given nutrient is provided at adequate levels, deficiencies can still arise from secondary factors, such as “nutrient lockout” in substrates, competition, antagonism for uptake with other elements, or suboptimal rootzone pH (Zheng, 2022) [3].

Prior studies have illustrated some of the potential impacts that suboptimal supply of nutrients can have on cannabis growth, yield, and secondary metabolite composition. Within normal sufficiency levels, nutrient supply does not appear to have substantial effects on inflorescence secondary metabolite composition (Bevan et al., 2021) [4]. However, relatively low or high nutrient supply levels have been shown to increase or reduce inflorescence secondary metabolite content, respectively (Caplan et al., 2017a; Saloner and Bernstein, 2022b; Shiponi and Bernstein 2021a) [5,6,7]. Therefore, there may be little commercial benefit of using fertility stress to manipulate secondary metabolite content due to the trade-off between increased secondary metabolites and reduced yield, commonly referred to as the “dilution effect” (Caplan et al., 2017a; Shiponi and Bernstein 2021a) [5,7].

Foliar tissue sufficiency ranges for individual nutrient elements are relatively well known for many commercially grown horticultural commodities; however, differences between and even within commodities can be substantial, depending on growing environments and other production inputs (Barker and Pilbeam, 2015) [8]. While some studies have reported nutrient sufficiency ranges in cannabis foliage grown in various production systems (e.g., Bernstein et al., 2019; Cockson et al., 2019; Kalinowski et al., 2020; Landis et al., 2019) [9,10,11,12], the optimum fertility levels for cannabis grown in different controlled environment systems and at different growth stages are still relatively undefined. Cannabis nutrition during the flowering stage is of particular importance due to its relatively long timespan and the complex biomass partitioning dynamics as plants transition from vegetative to generative growth and eventual senescence (Crispim Massuela et al., 2022; Potter, 2014) [13,14]. Furthermore, the concentrations of individual elements, both nutrient and non-nutrient, can affect the quality and marketability of harvested cannabis tissues. For example, it is commonly believed that flushing fertilizer nutrients from the rootzone during the final pre-harvest growth phase can enhance quality of marketable tissues (e.g., mature, unfertilized female inflorescences) (Caplan et al., 2022) [15]. Furthermore, some cannabis genotypes have been shown to hyper-accumulate both nutrient (e.g., Cu and Zn) and non-nutrient (e.g., Pb and Cd) heavy metals in inflorescence tissues (Angelova et al., 2004; Bengyella et al., 2022; Seleiman et al., 2012) [16,17,18]. Since heavy metals can be toxic to humans, the concentrations of heavy metals in marketed cannabis tissues are strictly controlled under most government regulations. Therefore, minimizing the presence of these elements in cannabis production systems (e.g., fertilizers, piping, growing substrates, etc.) is of utmost importance.

Despite some cannabis foliar tissue nutrient sufficiency ranges having been reported in the literature, high-quality images and accompanying information describing the onset and development of deficiency symptoms in cannabis is still relatively lacking. In many cases where images were provided, they were either of vegetative-stage cannabis plants (Cockson et al., 2019; Saloner and Bernstein, 2020; Saloner et al., 2019; Shiponi and Bernstein, 2021b) [10,19,20,21] or were taken at or near harvest maturity when natural senescence processes may confound the identification of specific foliar deficiency symptoms (Saloner and Bernstein, 2022a; Shiponi and Bernstein, 2021a; Saloner and Bernstein, 2021; Saloner and Bernstein, 2022b)) [6,7,22,23]. The quality of images in these articles is often insufficient for clear identification of deficiency symptomology, its location on the plant, or make comparisons among treatments. While there are abundant images and descriptions of cannabis nutrient deficiencies in industry publications and internet resources, few are supported by peer-reviewed research. Cockson et al. (2019) [10] is the only prior study that we are aware of that investigated the temporal development of different cannabis nutrient deficiencies. However, their nine-week trial was conducted on a hemp cultivar, and only during the vegetative growth phase. While this is certainly an important reference for cultivators, there remains general lack of high-quality pictures of nutrient deficiency symptoms generated and published based on scientific research on drug-type cannabis, especially during the flowering stage. These types of pictures are essential in guiding cultivators in diagnosing cannabis nutrient disorders in controlled environments. Furthermore, the analysis of foliar tissue nutrient composition in prior cannabis nutrient studies have generally focused on the most recently developed leaves, regardless of the elements of concern, and in many cases foliar samples were taken at or near harvest rather than at the onset of visual deficiency symptoms. These represent major knowledge gaps in cannabis fertility management.

To maximize plant health, yield, and quality, it is critical for cannabis cultivators to have tools to quickly and accurately determine potential deficiency conditions based on the early onset and development on foliar symptomology during the flowering stage. The main objective of this study was to use incomplete nutrient solutions to induce single-element nutrient deficiency symptoms in indoor-grown, drug-type cannabis at the start of the flowering stage and follow the onset and development of deficiency symptoms through to inflorescence maturity. The second objective was to evaluate nutrient element levels in foliar tissues with respect to the onset of visual deficiency symptoms. The third objective was to demonstrate the relative severity that deficiencies of different nutrients can have on cannabis yield and quality, including secondary metabolite composition.

Guidelines describing the onset and progression of foliar nutrient deficiency symptoms, supported by high quality images and corresponding tissue analyses, will assist cannabis cultivators in diagnosing nutrient disorders and taking appropriate corrective actions to minimize losses in yield and quality. Greater understanding of the relationships between nutrient levels in fertigation solutions and foliar tissues can also help cultivators make adjustments in nutrient supply before nutrient stresses result in serious consequences on plant productivity.

## 2. Materials and Methods

Cuttings were taken from a drug-type *Cannabis sativa* “Gelato 29” cultivar (HEXO Corp, Brantford, ON, Canada) on 9 March 2020, and rooted in 50 cell rockwool plug trays (AMA Horticulture Inc., Kingsville, ON, Canada). The trays were pre-hydrated with a nutrient solution, comprised of Dutch Nutrients Gro A and Gro B (Homegrown Hydroponics, Toronto, ON, Canada). The fertilizers were each prepared at a concentration of 5 mL·L^−1^ in reverse osmosis water, resulting in an EC of 1.7 dS·m^−1^ and pH of 5.8. The trays were placed under T5 fluorescent lights at a canopy photosynthetic photon flux density (PPFD) of approximately 200 µmol∙m^−2^∙s^−1^ and an 18 h photoperiod.

After 10 d of rooting, uniform rooted cuttings were planted into 6” net pots (0.62 L; FHD Plastics) filled with expanded clay pebbles (8–16 mm; Liapor, Hallerndorf, Germany), and inserted flush to the lid of 19-L (0.28 m diameter × 0.39 m height) black plastic buckets to make deep water culture (DWC) systems. Stakes and plastic support rings (Grower’s edge, Hawthorne Gardening Company, Surrey, BC, Canada) were affixed to the center of each DWC lid to provided plant support through the trial. Thirty-four DWC buckets were placed in a concealed grow room, with 1000 W metal halide lights as the sole source of photosynthetically active radiation (PAR) at an average canopy level PPFD of approximately 450 µmol∙m^−2^∙s^−1^ and an 18 h photoperiod. The day (i.e., when lights were on) and night temperature setpoints in the grow room were maintained at 24 °C and 20 °C, respectively.

Each DWC was continuously aerated with an air-stone (ASC030; Pawfly) attached to a compressed air line running at ≈0.5 L∙min^−1^. The maximum water level of each DWC was set to 17 L by drilling a 0.3 cm drain hole at 7.0 cm below the bucket’s upper rim. At transplant, the nutrient solution was comprised of Gro-A and Gro-B (Dutch Nutrients, Homegrown Hydroponics, Toronto, ON, Canada), each at a rate of 5 mL∙L^−1^ in rainwater, resulting in EC of 1.8 dS∙m^−1^ and pH of 5.8. The nutrient element concentrations (mg∙L^−1^) in the vegetative stage nutrient solution are presented in Table 1. The rainwater normally had EC ≤ 0.1 dS∙m^−1^ and was treated with hydrogen peroxide to 100 PPM at least 24 h before use. The plants were grown under metal halide lights on an 18 h daily photoperiod for 16 days.

### 2.1. Nutrient Deficiency Treatments

On day 16 after transplanting into the DWCs, the daily photoperiod was reduced to 12 h to provoke robust flowering responses. On the same day, the DWCs containing the 27 most uniform plants were spread into four rows of seven (less one plant in the fourth row) with 0.3 m spacing between adjacent DWCs and 0.6 m spacing between rows. Selected plants were (mean ± SE) 32 ± 0.7 cm tall and had 12 ± 0.2 nodes. The nutrient solution in each DWC was drained, then refilled with rainwater to rinse the roots. Each of the following nine nutrient treatments were assigned to three DWCs in a completely randomized design: control, -N, -P, -K, -Ca, -Mg, -S, -Fe, and -Mn, where the “-” denotes the missing nutrient element in each respective treatment. The DWCs were drained again and refilled with 17 L of nutrient solution from their respective treatments. The treatment nutrient solutions were made in 60 L batches using combinations of stock solutions in rainwater, with each stock solution comprised of a single reagent-grade salt dissolved in deionized water. Stock solution concentrations and the compositions of each treatment solution followed the methods of Barnes et al. (2012) [24] and are provided in Table 2. The calculated elemental concentrations of each of the treatment solutions are provided in Table 1. Elemental analysis of samples taken from the rainwater (i.e., source water), and freshly made solutions for each treatment were performed by an independent analytical laboratory (by SGS Canada Inc., Guelph, ON, Canada). The concentrations of N and Cl were determined using ion specific electrodes, and all other elements were determined using optical emission spectrometry (Table 3).

The nutrient solutions were drained and replaced with freshly made solution of their respective treatments on a weekly basis. In the interim periods, nutrient solution volumes were monitored daily and topped-up with rainwater on or before volumes reached 50%, meaning the maximum concentration of individual elements should not have exceeded 2× the calculated initial solution concentrations (Table 2).

Plants were grown for 53 days in the nutrient treatments, then harvested. The plants were examined daily and, starting in week two, digital photos of each plant (12.2 MP, ≈0.8 m focal distance) were taken weekly and at harvest. As deficiency symptoms progressed, photographs and scans (CanoScan LiDE 25, Canon Inc., Tokyo, Japan) of individual leaves were taken that were illustrative of deficiency progression. Observations of the onset and development of deficiency symptoms were carefully recorded. On each plant, at the onset of visual symptoms, ≈20 g samples were taken of leaves from the lower- and upper-canopy areas. On each plant that did not develop visual deficiency symptoms by the end of the trial, ≈20 g foliar samples were taken from both lower and upper canopy sections at harvest. Leaf tissues were dried at 60 °C to constant weight and submitted to an independent laboratory (A&L Canada Laboratories Inc., London, ON, Canada) for analysis of foliar nutrient composition using the methods described in Yep and Zheng (2021) [25].

### 2.2. Harvest and Postharvest Analysis

At maturity, plants were harvested individually (i.e., one at a time) in a random order. Each plant was divided into inflorescence (trimmed of sugar leaves according to normal commercial practice), aboveground vegetative tissue (stems and leaves), and roots. The fresh weight (FW) of the aboveground tissues were weighed separately immediately after dividing the tissues. Root balls were shaken to remove most of the free water and left to dry under the 1000 W metal halide lights for 1 week, rotating each root ball every other day, then oven dried at 60 °C to constant weight before the dry weights (DW) were recorded. The apical inflorescence from each plant was dried on perforated drying trays at (mean ± SD) 19 ± 1.8 °C and 51 ± 9.2 RH% for 5 d (final moisture content of ≈12%). Drying room air temperature and RH were logged every 5 min (HOBO MX1102A, Onset). After air drying, the inflorescence material was homogenized, and ≈2 g samples from each plant were submitted to a Health Canada approved lab (RPC, Fredericton, NB, Canada) for analysis of cannabinoids. Using proprietary internally developed methods and standards, the concentrations (mg∙g^−1^ of dry tissue) of the following cannabinoids were quantified using solvent extraction by ultra-high-performance liquid chromatography with variable wavelength detection: cannabichromene (CBC), cannabidiol (CBD), cannabidiolic acid (CBDA), cannabigerol (CBG), cannabigerolic acid (CBGA), cannabinol (CBN), Δ^8^-tetrahydrocannabinol (d8THC), Δ^9^-tetrahydrocannabinol (THC), and Δ^9^-tetrahydrocannabinolic acid (THCA). Total equivalent cannabidiol (T-CBD) and total equivalent Δ9-tetrahydrocannabinol (T-THC) were calculated by assuming complete carboxylation of the acid-forms of the respective cannabinoids and factoring out the acid moiety from the molecular weights of the acid forms of each respective cannabinoid (e.g., T-THC = (THCA × 0.877) + THC].

### 2.3. Descriptions of Deficiency Symptoms

Descriptions of foliar symptoms will follow the terminology for cannabis plant morphology outlined in Cockson et al. (2019) [10] combined with deficiency symptomology used in Resh (2012) [2]. Foliar positions on the plant will be described as either lower (i.e., older), middle or upper by dividing the plants into thirds by height. Furthermore, foliage will be subdivided into the following two groups: fan and sugar. Fan leaves are the predominant leaf type that develop during the vegetative growth phase (Raman et al., 2017) [26] which normally approximately 5 weeks beyond the switch to a 12 h photoperiod (Potter, 2014) [14]. Sugar leaves are smaller and are closely associated with developing inflorescence tissues, often producing high densities of glandular trichomes, especially on the basal portion of the leaflets. Generally, the age of fan leaves progress from older to younger from the bottom to the top of the plant and from lower- to higher-order branches. Many of the fan leaves are already present before sugar leaves begin to appear. Therefore, fan leaves are generally older than sugar leaves, especially leaves at the bottom parts of the plants and attached to the main stem (Bernstein et al., 2019; Raman et al., 2017) [9,26].

### 2.4. Experimental Design and Statistical Analysis

The experiment was conducted as a completely randomized design with nine treatments and three replications. The trial ran for 7.5 weeks under 12-h days, at which point inflorescences had developed conventional signs of commercial maturity (i.e., when plants are normally harvested to optimize yield and secondary metabolite composition), including stigma browning and stalked trichomes that had transitioned from transparent to a milky translucent appearance. Analysis of harvest and postharvest metrics were done using JMP (version 10; SAS Institute Inc., Cary, NC, USA) with means separation between treatments using Tukey’s honestly significant difference test at a significance level of *p* ≤ 0.05. Intra-treatment analysis of lower- vs. upper-canopy foliar elemental concentrations were also done in JMP on each element using Tukey’s honestly significant difference test at a significance level of *p* ≤ 0.05.

## 3. Results and Discussion

This trial deliberately induced nutrient deficiency symptoms by withholding individual nutrient elements from respective treatment solutions while maintaining sufficient levels of all other nutrient elements. While complete absence of any single nutrient element is unlikely to occur in a commercial production system, it is important to do this experimentally to positively establish cause-effect relationships between the deficient element and associated symptoms. One of the challenges in the induction of single-element nutrient deficiencies is that plant fertilizers are predominantly comprised of mineral salts (i.e., ionic compounds that disassociate into positive- and negative-charged ions when dissolved in water). Consequently, it is not easy to withhold single nutrient elements from a conventional fertilizer recipe without also affecting the counter-charged ion (in any given salt), possibly upsetting the balance of other nutrients or limiters. Therefore, transparency on the individual chemicals in nutrient recipes and resulting elemental concentrations in nutrient solution, substrate matrices (if present), and plant tissues are important information to disclose. For example, some of the differences in symptomology between the present study (described below) and some cited works may be partially attributed to uncertainties in the nutrient composition of the respective fertilizer recipes. Furthermore, foliar symptoms attributed to a specific nutrient deficiency may be partially confounded with natural foliar senescence at the end of cannabis’ normal ontology.

In the present study, there were elevated levels of Na and Cl in some treatments because Na- and Cl-based salts were used to replace other common salts. For example, calcium nitrate was the major source of N in every treatment except for -Ca, where sodium nitrate was substituted, hence the higher Na concentration in the -Ca treatment. In all cases, the Na and Cl concentrations in fresh treatment solutions were below those normally considered to be limiting (Zheng, 2022; Yep et al., 2020a;) [3,27]. Except for the deficient nutrient in each respective treatment which were at or below the levels measured in the source-water, the concentrations of all other nutrient elements were at similar levels as the control treatment (Table 3). Additionally, note that these were starting concentrations, which may have as much as doubled from time to time, according to the weekly DWC refilling protocol. Therefore, the observed foliar symptomology could be expected to exemplify the onset and progression of deficiency symptoms associated with the most rapid stage of vegetative growth (i.e., at the start of the 12/12 h flower promoting photoperiod). The upper and lower canopy foliar tissue samples were collected during week four in -N, -K, and -Mg; week five in -S; week six in Control, -P, and -Ca; and week seven for -Fe and -Mn treatments, respectively. According to the foliar analysis report, Cl was not analysed, and the concentrations of Zn, Cu, B, and Al were within the “sufficient” to “very high” ranges in all samples (data not shown).

### 3.1. Onset and Progression of Visible Deficiency Symptoms by Nutrient Elements

The temporal progressions of foliar nutrient deficiency symptoms for the control and each deficient treatment are detailed in the following sections. Full plant images of each treatment in each week shows the impact of the nutrient deficiencies on the overall plant morphology (Appendix A). Accompanying images of the apical inflorescences from each nutrient deficiency treatment at harvest are presented in Appendix A.

#### 3.1.1. Control

The plants in the control treatment illustrate the normal appearance of cannabis development as it transitions from vegetative stage to generative growth, to inflorescence maturation, to early senescence. Throughout the first five weeks of the flowering stage, control plants had no visible foliar nutrient deficiency symptoms (Figure 1a). However, by week six there was moderate chlorosis on the oldest (largest) fan leaves attached to the main stem (Figure 1b), which became more severe during the following weeks (Figure 1c,d). The chlorosis generally increased from bottom to top of the plant and from leaflet edges towards the midrib. For a given fan leaf, the severity of chlorosis generally increased from the central leaflet towards the smaller, radial leaflets (Figure 1c,d). These chlorosis symptoms appeared, despite there being no deficient elements in either the nutrient solution (Table 1 and Table 3) or the sampled foliar tissue (Table 4). These leaf senescence phenomena are normally observed in maturing cannabis plants even when nutrients are supplied at luxury levels (based on personal observations and communications with growers). It is important to be mindful of the normal senescence phenomena that develop as indoor-grown cannabis plants approach harvest maturity. This is because these natural processes can be mistaken for signs of abiotic stresses, such as nutrient deficiencies. For example, natural senescence may be confounding the foliar discoloration which have been attributed to nutrient disorders in some of the whole plant images presented in prior studies (e.g., Saloner and Bernstein 2022a; Shiponi and Bernstein, 2021a; Saloner and Bernstein 2021; Saloner and Bernstein 2022b; Yep et al. 2020b) [6,7,22,23,28]. Furthermore, the common commercial practice of “flushing” nutrients by withholding fertilizers in the last few weeks before harvest (Caplan et al., 2022) [15] may amplify the appearance of nutrient deficiency symptoms or accelerate natural foliar senescence. Cultivators must be extra cautious when assessing foliar disorders in the latter stages of cannabis flowering.

The normal development of the apical inflorescence over the final five weeks before harvest (i.e., week three to week seven) is presented in Appendix A. These weekly images are illustrative of the profound increases in inflorescence biomass that occurs in the latter stages of production (Potter, 2014) [14]. The latter-week images also demonstrate the onset and development of foliar senescence which occurs naturally as plants approach the end of their life cycle (Resh, 2012) [2]. Discolorations associated with natural senescence, occurred even when the foliar nutrient concentrations of control plants (measured in week six, Table 4) were similar to sufficiency ranges reported by others (Bernstein et al., 2019; Cockson et al., 2019; Landis et al., 2019; Yep and Zheng, 2021) [9,10,12,25]. This highlights the challenges of attributing foliar discoloration in the latter stages of cannabis ontogeny to potential nutrient disorders.

#### 3.1.2. Nitrogen (N)

The first sign of deficiency in the -N plants was somewhat stunted vegetative growth relative to the other treatments. While the relative size of the plants in each treatment were not investigated quantitatively, reductions in size (e.g., increases in height and width) in the -N plants relative to most other treatments during the first few weeks (i.e., during the phase of rapid vegetative growth) were quite apparent. These observations are consistent with vegetative-stage cannabis grown at low levels of N-fertilization (Cockson et al., 2019; Saloner and Bernstein, 2020; Caplan et al., 2017b;) [10,19,29].

Visual symptoms of N deficiency began as very slight yellowing of the leaflet tips of lower canopy fan leaves during week three (Figure 2a). By the following week, yellowing was apparent on most of the fan leaves on the bottom two-thirds of the plants (Figure 2b). Severity of chlorosis of the fan leaves also progressed from the base of the leaflet to the tip and was initially most intense in the interveinal regions, followed by broad yellowing of all foliar tissue (Figure 2c). By week five, foliar yellowing was evident on almost all fan leaves with the level severity increasing from the bottom to the top of the plant. Petioles on the most affected leaves also turned from dull green to red-brown (Figure 2c). By week five, some fan leaves had turned brown and senesced, and the majority of the remaining fan leaves were very yellow (Figure 2d). The development of N deficiency symptoms on fan leaves were consistent with the descriptions of N-deficient vegetative-stage cannabis in Cockson et al. (2019) [10], late flowering stage cannabis in Saloner and Bernstein (2021) [22], and were generally similar N-deficiency symptomology in other crops (Resh, 2012) [2].

By week five, sugar leaves had also started yellowing, following the same patterns of increased yellowing from leaflet base to tip and from the bottom to the top of the plant. The main stem and stems of lower-order branches also began yellowing. By week six, almost all of the fan leaves had senesced, and sugar leaf yellowing appeared to be progressing starting at the older, larger leaves at the bases of inflorescences. Sugar leaves on lower levels of the plant or higher order branches appeared to be more affected than at the distal end of the main stem (i.e., the cola) and apices of side branches. The progression of N deficiency symptomology on sugar leaves during the latter weeks can be seen in Appendix A**.** By week seven, the majority of sugar leaves were yellow-brown with only the highest-order leaves remaining somewhat green. All stem tissues had turned brown by the end of the trial. However, this added information may be of little commercial relevance since crops exposed to uncorrected severe N-deficiency for this long (i.e., ≥ 5 weeks) would be accompanied by substantial losses in yield and appearance quality of the mature inflorescences (Table 5, Appendix A).

There were no differences in upper and lower canopy foliar N concentrations at the onset of deficiency symptoms, and they were similar to other treatments including the control (Table 4). Furthermore, while the foliar N concentrations were somewhat lower than the sufficiency ranges reported in Landis et al. (2019) [12], they were similar to the commercial fertilizer (i.e., control treatment) and various fertilizer supplementation treatments in Bernstein et al. (2019) [9], and in week five of flowering in solution culture containing moderate levels of N (Yep et al., 2020b) [28]. This is illustrative of the potential limitations of using foliar tissue analyses to evaluate the onset or development of nutrient deficiencies, especially later in the production cycle.

Relative to the control treatment, the substantial reductions in yield in the N-deficiency treatment were not accompanied by treatment effects on cannabinoid concentrations (Table 6), which contrasts with some prior studies. Saloner and Bernstein (2021) [22] and Bevan et al. (2021) [4] showed the importance of N supply on cannabis growth, yield and quality, with economic optimums probably in the 150 to 200 mg·L^−1^ range. However, there were inconsistent treatment effects on cannabinoid concentrations in these studies. In addition, Saloner and Bernstein (2022b) [23] highlighted the importance of using the right source of N [e.g., nitrate (NO_3_) vs. ammonium (NH_4_)] in cannabis fertilizer. They found that yield and quality generally decreased with increasing NH_4_:NO_3_. While this may have limited practical value in assessing the onset of N deficiencies, it does illustrate the economic importance of maintaining both N supply and composition within appropriate levels for optimum cannabis growth and yield.

Cultivators should also be mindful that N is one of the highest demand nutrients with some of the highest tissue concentrations of any mineral nutrients (Cockson et al., 2019; Landis et al., 2019) [10,12]. Furthermore, symptoms of N deficiency tend to develop more rapidly than some other nutrients, as observed in this trial.

#### 3.1.3. Phosphorous (P)

Foliar symptoms of phosphorous deficiency did not begin to appear until almost halfway through the flowering stage. By week three, some lower fan leaves began to show tiny chlorotic spots (Figure 3a). By week four, these spots had increased in size, and some began to coalesce into larger necrotic lesions (Figure 3b). This symptom has not previously been described in indoor grown cannabis (e.g., Shiponi and Bernstein, 2021a; Cockson et al., 2019; Shiponi and Bernstein, 2021b) [7,10,21] and appears to be uncommon in other plant species. However, this symptom only appeared in the -P treatment, indicating it was a symptom of P-deficiency. By week five, the necrotic spots in more severely affected leaves had amalgamated into larger, irregularly shaped necrotic areas along with some mild general yellowing of the unaffected areas (Figure 3c). The tips of many of the leaflet dentations had also become necrotic. The various stages of the development of phosphorous deficiency symptoms on fan leaves, from small spots to large necrotic, are shown in Figure 3d. In some cases, the leaf margin necrosis, also described in Cockson et al. (2019) [10], resembled injury often associated with waterlogging (Zheng, 2022) [3].

In some cases, the necrosis of leaf dentations formed a continuous ring of necrotic tissue around the entire leaflet margin. By week six, the most severely affected fan leaves had become necrotic, and the leaflets curled upwards, both laterally (side to side) and longitudinally (tip curling backwards) (Figure 3e). By week six, necrotic areas also began developing on larger sugar leaves, following similar patterns as the development on fan leaves. By week seven, all fan leaves had died, and many of the larger sugar leaves developed necrotic areas, particularly along the margins and tips of the leaflets (Figure 3e, inset). Some tip browning appeared on some of the smaller sugar leaves. The roots of the plants in the phosphorous deficiency treatment appeared to develop faster than other treatments and had the highest DW at harvest (Table 5).

Notably, other than moderate purpling in leaf petioles (Figure 3d), there was little evidence of enhanced anthocyanin expression, which is a common symptom of P deficiency in other crops (Resh, 2012) [2] which is often noted as purpling, especially on abaxial leaf surfaces. The lack of foliar purpling under P-deficiency was consistent with the observations of vegetative (Shiponi and Bernstein, 2021b) [21] and flowering (Shiponi and Bernstein, 2021a) [7] cannabis grown in low P treatments.

The plants in the -P treatment had the second lowest aboveground biomass, but also the most prolific root growth of all treatments (Table 5), which is contrary to the reduced root development in the lower P treatments in both vegetative- and flowering-stage cannabis (Shiponi and Bernstein, 2021a,b) [7,21] and other species grown under P-deficiency conditions (Resh, 2012) [2]. The inflorescence THC concentration was the highest in the -P treatment plants but was only significantly higher than plants in the -Ca treatment (Table 6). Shiponi and Bernstein (2021a) [7] also reported that reduced P supply enhanced THC concentrations in both primary and secondary inflorescences. However, this was overshadowed by greatly diminished THC yield (mg/plant), due to reduced inflorescence biomass. Overall, withholding P may have some potential as a production tool to encourage early cannabis root development, but there would be little production value in withholding P after flowers appear to enhance THC concentrations.

#### 3.1.4. Potassium (K)

There were no visible potassium deficiency symptoms for the first two weeks of the trial. However, by week three, the foliage of potassium-deficient plants began to display two disparate types of symptoms, depending on the type of leaf. Most obvious were dark brown lesions surrounding the secondary branch veins of larger sugar leaves (Figure 4a). Conversely, lower fan leaves showed light browning, predominantly at the leaflet tips and along the dentations of the leaf margins. The symptoms on the fan leaves progressively transitioned from yellowing to browning, and started to develop one week later (i.e., week four) than the potassium deficiency symptoms on the sugar leaves (Figure 4b).

The drying and scorching of leaf tips and margins of fan leaves observed during this trial are commonly observed on other K-deficient terrestrial plant species (Resh, 2012) [2], but our results contrast with other cannabis studies. Furthermore, while some leaf margin scorching was visible in the lowest K fertilizer treatment in flowering cannabis, this was a minor symptom compared to the generalized whole plant chlorosis observed in both vegetative and flowering cannabis plants grown under low K fertility (Saloner and Bernstein, 2022a; Saloner et al., 2019) [6,20]. Additionally, it is notable that while the foliar K concentrations (at the onset of symptoms) were about half of the other deficiency treatments, there were no differences in upper vs. lower canopy K concentrations in the -K treatment (Table 4). The foliar K concentrations in the present study were also substantially higher than in the K-deficiency treatment in Cockson et al. (2019) [10] or in the reduced K-treatments in Saloner and Bernstein (2022a) [6] and Saloner et al. (2019) [20].

By week five and beyond, both of these disparate deficiency symptoms were visible on the same plant, sugar leaf symptoms were predominant at the top of the plant and fan leaf symptoms were most intense in the middle third (Figure 4c,d). The light brown coloration on the fan leaves progressively extended inwards on the leaflets, particularly in the interveinal regions. By week six, the only green tissues that remained were at the inflorescences, along the midrib of sugar leaves, and some of the lower fan leaves. By week seven, most non-inflorescence leaves were entirely brown, curled, and dried out. Some inflorescence leaves began to show tip browning as well.

Plants in the -K treatment had the second lowest root dry weights of all treatments and less than half of the inflorescence fresh weight as the control plants (Table 5). However, inflorescences in the -K treatment also had the highest THCA and GBGA (the precursor to THCA and CBDA) concentrations (Table 6). These results were consistent with the results in Saloner and Bernstein (2022a) [6] where lower K supply promoted higher concentrations of these cannabinoids, at the cost of reduced inflorescence yield.

#### 3.1.5. Calcium (Ca)

By week three, lower fan leaves began developing spots between the veins, which were comprised of yellow rings around necrotic brown centers (Figure 5a). By week four, most affected leaves were in the bottom two-thirds of the plants, with severity generally decreasing with height. Spots were present over the entire leaf surface, but they appeared to be concentrated near the leaf edges and the leaflet tips (Figure 5b) such that by week five they had coalesced on the most affected leaves to form irregular necrotic areas along the margins (Figure 5c). Many leaf margin dentations had also turned brown. As Figure 5. Progression of calcium (Ca) deficiency, (a) starting on lower fan leaves in week three, with severity of symptoms being concentrated near leaf margins (b). By week five there was general leaf margin necrosis on lower fan leaves (c) an upper fan and sugar leaves started to show symptoms (d). By week six, the majority of fan and sugar leaves showed characteristic leaf margin necrosis and upper leaves were curled upwards (e).As time passed, the leaf spotting moved up the plant to the smaller, younger fan leaves and leaf tips of upper fan leaves also started curling upwards (Figure 5d). In week five, some of fan leaves at the very bottom of the canopy also started showing generalized chlorosis, which decreased in severity from leaflet tip to base—these symptoms became more pronounced in the following week but remained localized in the lower canopy fan leaves (Appendix A). By week six, the severity of edge necrosis and upward curling of fan leaves had spread across the entire plant, and larger sugar leaves at the top of the plant had also begun developing similar symptoms. The development and progression of Ca-deficiency symptoms were markedly different from those described in vegetative-stage cannabis (Cockson et al., 2019) [10]. Since Ca is normally considered an immobile nutrient, these observed symptoms are also a departure from typical calcium deficiency in terrestrial plants, where foliar symptoms normally start in younger tissues (Resh, 2012) [2]. Despite the observed symptomology, the foliage in the lower-canopy did have almost three times higher Ca concentrations than the upper canopy at the onset of symptoms (Table 4).

The upper and lower canopy Ca concentrations in the -Ca treatment were within or greater than the sufficiency ranges healthy-looking, vegetative-stage plants from five greenhouse grown cannabis cultivars reported in Landis et al. (2019) [12]. Since the -Ca treatment had the highest solution and foliar Na concentrations and also the highest foliar K and Mg levels, the -Ca treatment solution may have led to imbalances in nutrient uptake, and the development of Ca-deficiency symptomology.

Ca deficiency had moderate effects on biomass production, including inflorescence yield (Table 5), but Ca-deficient plants had the lowest cannabinoid concentrations of all of the treatments (Table 6), possibly partly due to the high Na levels in this treatment (Yep et al. 2020a) [27].

#### 3.1.6. Magnesium (Mg)

The were no visible deficiency symptoms in the magnesium-deficient plants for the first three weeks. In week four, mild interveinal chlorosis began to develop in lower fan leaves (Figure 6a). The interveinal chlorosis appeared first in regions between the midrib and the leaf margins. Magnesium deficiency symptoms developed quickly beyond week four, predominantly affecting smaller, younger fan leaves (Figure 6b) whereas older fan leaves and sugar leaves within inflorescences were less affected. The leaves most severely affected by magnesium deficiency started to develop interveinal necrotic regions, with severity increasing from leaflet bases to tips (Figure 6c). By week five, magnesium deficiency was widespread over the entire plant, with the apical inflorescences being least affected (Figure 6d). By week six, most of the smaller fan leaves and sugar leaves showed severe yellowing in all areas except for the midrib and extreme leaf margins, while some older fan leaves remained largely unaffected (Figure 6e).

Interveinal chlorosis of older leaves is a common and distinguishing symptom of magnesium deficiency in terrestrial plants (Resh, 2012) [2] and similar foliar symptomology for Mg deficiency was reported in vegetative-stage cannabis (Cockson et al., 2019) [10]. When Mg deficiency symptoms first appeared, lower-canopy foliar Mg concentrations were similar to the foliar Mg concentrations in Mg-deficient cannabis in Cockson et al. (2019) [10] and three times lower than the upper canopy foliage. Cockson et al. (2019) [10] reported no effects of Mg deficiency on vegetative plant biomass relative to their control treatment, which was similar to the present study. However, although there were no treatment effects on cannabinoid concentrations, inflorescence yields in the -Mg treatment were approximately 30% lower compared to the control treatment. Perhaps the role of Mg in cannabis growth may be disproportionately important for the production of floral biomass.

#### 3.1.7. Sulfur (S)

Symptoms of sulfur deficiency first appeared during week four, as a general yellowing on the recently developed upper sugar leaves (Figure 7a). By week five, upper fan leaves had a pronounced upward angle (Figure 7b), and lower fan leaves started developing interveinal chlorosis, which extended from the midrib all the way out to the leaf margins (Figure 7c). The general chlorosis of upper sugar leaves that had first appeared in week four did not appear to progress over time, except for browning on the leaf tips and the margin dentations. Most of the later sulfur deficiency symptoms developed in the fan leaves similar to the foliar chlorosis first seen in week five (Figure 7d); however, these symptoms progressed to total foliar yellowing and eventual senescence. There were no significant differences in upper- vs. lower-canopy foliar S concentrations, but the foliage in the S-deficient plants had substantially lower S content than in all other treatments, including the control.

Overall, the leaf chlorosis symptoms caused by S deficiency were similar to the symptoms caused by N deficiency; however, symptoms began in the upper-canopy foliage, and it took relatively longer for S deficiency to be evident. The location of first appearance of symptoms somewhat contrasted with vegetative-stage cannabis in Cockson et al. (2019) [10] [i.e., in upper-canopy foliage (sugar leaves) vs. the “middle of the plant”]. However, while the locations (on the plant) of deficient tissues are not clear in their photo, the S-deficiency symptoms appeared to be most pronounced near the localized growing points (i.e., branch apices), regardless of plant position. Furthermore, the progression of symptomology in the present study was from midrib outwards and from leaf tip to base vs. from base to tip described in Cockson et al. (2019) [10].

Sulfur deficiency had no major impacts on vegetative and root biomass or cannabinoid concentrations, but inflorescence yield was reduced by 34% vs. the control plants.

#### 3.1.8. Iron (Fe)

There were no striking deficiency symptoms that developed in the -Fe treatment during this trial. In early flowering, the sugar leaves and flowers appeared a paler green than the same tissues in other treatments (Figure 8a). By week five, some of the smaller young leaves associated with the apical inflorescences on the main stem and lower-order branches developed some mottling, whereby the middle portions of some leaflets were lighter green, showing chlorosis, than either the bases or tips (Figure 8b). However, this mottling of the foliar tissues associated with apical inflorescences, seemed to disappear by later in the trial (Figure 8c). By week six, many of the lower fan leaf margins had turned light brown (Figure 8d).

Foliar Fe concentrations at harvest in -Fe treatment plants were similar to the -Mg and -Ca treatment and lower than the rest of the treatments. While this was not statistically significant, lower canopy Fe concentrations were generally higher than upper canopy tissues. Since Fe is considered a relatively immobile element, the appearance of Fe deficiency symptoms in younger tissues first is consistent with most terrestrial species (Resh, 2012) [2]. The plants in the -Fe treatment had some of the highest foliar concentrations of some other elements, including Ca, S, Mn, Zn, and Al, suggesting that Fe deficiency may provoke hyper-accumulation of other cationic elements. Fe deficiency had no deleterious effects on plant biomass and or on the concentrations of major cannabinoids.

#### 3.1.9. Manganese (Mn)

No visually obvious signs of manganese deficiency were apparent in the -Mn treatment throughout the trial. By the end of the trial, the plants in the-Mn treatment (Figure 9a) had similar appearance to the control treatment (Figure 9b), and apical inflorescences of both treatments were similar in size, shape, and color. There were no treatment effects on the biomass or cannabinoid concentrations of the -Mn plants. At harvest, the upper canopy foliar Mn concentrations in the -Mn plants were extremely low compared to the lower canopy foliage and the foliar Mn concentrations in all other treatments, similar to Cockson et al. (2019) [10]. Given the very low upper-canopy foliar Mn concentrations in the -Mn treatment plants, without any deficiency symptoms or decreases in biomass (similar to Cockson et al., 2019) [10] perhaps cannabis is especially efficient at up taking and utilizing Mn when it is scarce. On the other hand, cannabis may be quite sensitive to Mn toxicity (Zheng, 2022; Cockson et al., 2019; [3,10].

### 3.2. Dynamics of Foliar Tissue Nutrient Compositions

Overall, the elemental analysis of foliar tissues in each treatment showed varying responses to the respective nutrient deficiencies (Table 4). There were no differences between upper and lower canopy foliar concentrations of the N, P, K, S, and Fe in their respective deficiency treatments; however, the concentrations were generally lower than in the other treatments, including the control. The concentrations of Ca and Mn were highest in the lower canopy foliage, and Mg was highest in the upper canopy foliage, in their respective deficiency treatments. In addition to the target deficient elements, there were many occurrences of differences between upper and lower canopy elemental concentrations in nutrients that were not the deficient element within the respective deficiency treatments. Only the -K and -Mg treatment did not have any upper vs. lower canopy differences in all elements that were provided at sufficiency levels. A temporal sampling strategy (e.g., weekly) and/or collecting samples from a greater number of plants (i.e., higher “n”) may decrease the inherent uncertainty levels in foliar tissue analyses and provide more insight on potential nutrient interactions in nutrient deficiency treatments.

### 3.3. Limitations of Nutrient Deficiency Trials and the Need for Integrated Approaches to Diagnosing Plant Nutrient Disorders

The foliar symptomology of the different nutrient deficiencies of flowering-stage cannabis plants in the present study sometimes emulated and sometimes contrasted with symptoms normally observed in other terrestrial plants, including specifically in cannabis. There may be several reasons for this.

It is possible that there are some unique features of indoor-grown cannabis and its cultivation environment that may lead to some atypical responses to nutrient deficiencies. For example, under optimized environmental conditions, such as much higher light intensities than most indoor-grown crops are exposed to, indoor-grown cannabis exhibits very high transpiration rates and prolific growth, possibly altering normal ranges of source-sink dynamics of nutrient allocation between different tissue types and locations on the plant. In such a case, an element that is normally considered to be “mobile” may not be re-mobilized fast enough from older tissues to fully support rapid new growth. An example of this might have been the observed sugar leaf symptomology that developed in concert with very different fan leaf symptomology in the -K treatment plants. In addition, the complete nutrient recipe in the vegetative stage fertigation solution (Table 1) may have provided adequate supplies of some of the lower-demand nutrients to support the plants’ needs through their entire flowering stage. This might explain why no visible deficiency symptoms occurred in the -Fe and -Mn treatments, even though the foliar levels of the respective deficient elements were less than half of the levels in the respective leaves in control treatment.

The elemental analysis of foliar tissues is an important production tool for identifying and characterizing potential nutrient deficiencies. The cannabis nutrient sufficiency ranges described in prior works provide valuable background information, although there can be substantial variability both among different cultivars and even for the same cultivar gown in different environments. Furthermore, when and where tissue samples are collected can have dramatic impacts on their nutrient content; readers must be cognizant of how the reported concentrations relate to the respective growing environments (including fertility levels), and when and where tissue samples were collected. Cannabis tissue nutrient concentrations vary in time, generally reducing as plants approach reproductive maturity (Dorais and Lelanc, 2022) [30], so sufficiency ranges based on foliar nutrient concentrations at harvest may not accurately reflect earlier stages of growth. The best strategy for identifying nutrient deficiencies is to collect foliar samples at or very soon after the onset of symptoms because nutrient concentrations in tissue samples that are collected weeks after the onset of deficiency symptoms are of little practical value in crop husbandry. Since plants can remobilize some nutrients from older to younger tissues more easily than others, analyzing individual tissue samples from both the upper and lower canopy may help to identify some nutrient deficiencies. An important benefit of analyzing foliar tissue for nutrient elements over the analysis of growing media or fertigation solutions for determination of nutrient deficiencies is that they show what has actually been absorbed vs. what may be available in the rootzone (Zheng, 2022) [3].

In some cases, plants exposed to certain nutrient deficiencies can maintain sufficiency levels in their foliage but have restricted growth (Zheng et al., 2018) [31], which can be hard to identify in commercial production systems. Cockson et al. (2019) [10] observed restricted early vegetative growth in their N-, P-, and Ca-deficiency treatments; plants in N- and P-deficiency treatments in the present study may have exhibited similar early stunting (Appendix A). Overall, even with careful selection of target foliar tissues and sampling time, it is not always possible to definitively diagnose nutrient deficiencies by relying solely on foliar tissue analysis. In fact, without additional supporting information, the foliar analyses from the -N treatment plants in the present study may be more suggestive of deficiencies of Ca or Mg rather than N, referencing the sufficiency ranges in Cockson et al. (2019) [10]. The best practice for evaluating any potential nutrient deficiency in a cropping system is to holistically consider all of the available information; including growing media nutrient composition and availability, elemental analysis of fertigation solutions and leachate (if available), foliar symptomology, and elemental analysis of foliar tissues that are sampled at strategic times and locations on affected and unaffected plants. Only with this integrated approach can experienced cultivators make well-informed decisions on potential nutrient disorders and take appropriate corrective actions.

### 3.4. Nutrient Deficiencies Reduced Yield but Had Only Minor Effects on Secondary Metabolite Composition

It has been well established that suboptimal supply of major nutrient elements during the flowering-stage can substantially reduce cannabis vegetative growth and inflorescence biomass (Bevan et al., 2021; Caplan et al., 2017a; Saloner and Bernstein, 2022a; Shiponi and Bernstein 2021a; Saloner and Bernstein, 2022b;) [4,5,6,7,23]. The deliberate induction of deficiencies in the present study may be illustrative of the relative magnitude of the impacts that individual nutrients can have on cannabis growth and yield. The vegetative and inflorescence biomass in -N and -P treatment were almost three times lower, and between 30% to 50% lower in the -K, -Ca, -Mg, and -S treatments vs. the control treatment (Table 5). Only the -Fe and -Mn treatment plants had no reductions in aboveground biomass, compared to the control treatment, none of which showed major signs of nutrient deficiency at harvest. The N- and K-deficient plants had the lowest root dry weights while the P-deficient plants had the highest root dry weights. There were also no nutrient deficiency treatment effects on harvest index, which was unexpected since deficiencies in less mobile nutrients (e.g., Ca and S) should have proportionally greater effects on the development of the relatively younger inflorescence tissues. Furthermore, the apportioning of individual nutrients between vegetative and inflorescence cannabis tissues have been shown to be quite variable (Saloner and Bernstein, 2022a; Shiponi and Bernstein, 2021a; Bernstein et al., 2019b; Saloner and Bernstein, 2022b; Kleinhenz et al., 2020) [6,7,9,23,32], there may be complex synergistic effects amongst individual nutrient elements (Saloner and Bernstein, 2021) [22], and in some cases nutrient supply appears to affect harvest index (Saloner and Bernstein, 2022a; Shiponi and Bernstein, 2021a; Saloner and Bernstein, 2022b) [6,7,23].

The “Gelato 29” cannabis cultivar is strongly THC-dominant with total THC comprising over 96% of the total cannabinoid concentrations (i.e., in the control treatment, Table 6). Compared to the nutrient treatment effects on biomass, there were relatively small treatment effects on cannabinoid concentrations, predominantly in the minor cannabinoids. Of particular note were the differences between the plants in the -K vs. the -Ca treatments. Generally, the -K treatment plants had amongst the highest concentrations of all measured cannabinoids, whereas the -Ca treatment plants had the lowest concentrations of some individual cannabinoids, including CBDA, CBGA, and THCA, which were each between 35% and 45% lower than in the -K treatment. Within the ranges of N (70–290 mg∙L^−1^), P (20–100 mg∙L^−1^), and K (60–340 mg∙L^−1^) supply. Bevan et al. (2021) [4] showed similar inflorescence concentrations of the major cannabinoids, but no nutrient treatment effects on cannabinoid composition in the same cultivar. Some other studies have shown that nutrient availability can affect inflorescence cannabinoid content (Caplan et al., 2017a; Saloner and Bernstein, 2022a; Shiponi and Bernstein, 2021a; Bernstein et al., 2019;; Saloner and Bernstein, 2021; Saloner and Bernstein, 2022b; Yep and Zheng 2021) [5,6,7,9,22,23,25]. Higher nutrient supply levels have sometimes been associated with decreases in some cannabinoid concentrations, but these were generally offset by relatively much greater increases in inflorescence yield (i.e., the so-called “dilution effect”) (Caplan et al., 2017a; Shiponi and Bernstein, 2021a; Yep et al., 2020b) [5,7,28]. Therefore, with inflorescence yield being the dominant economic driver in cannabis cultivation, any small increases in cannabinoid concentrations from growing plants in low nutrient conditions are unlikely to be commercially relevant. Within moderate levels of nutrient supply, nutrient concentrations may not substantially affect secondary metabolite composition but extreme nutrient supply levels (i.e., too low or too high) can. Whether nutrient stress can be used as a production tool to manipulate cannabis quality requires further investigation.

## 4. Conclusions

Due to its prolific growth under ideal conditions, indoor-grown cannabis is a high nutrient feeder. Nutrient imbalances, especially during the flowering-stage, have the potential to substantially alter growth and reduce inflorescence yield and quality. This study induced single-element nutrient deficiencies and combined observations on initiation and development of single-element nutrient deficiencies and analyses of nutrient concentrations in concurrently sampled upper- and lower-canopy foliage and fertigation solutions. Some of the individual nutrient deficiencies showed symptomologies that were common in terrestrial plants, including cannabis. However, there were some symptoms that were atypical, and may be specific to cannabis, particularly its high intensity cultivation in controlled environments. Individual nutrient deficiencies had substantial but varying effects on growth of vegetative and reproductive tissues (i.e., inflorescence yield), but relatively small effects on cannabinoid composition. Cultivators are urged to use an integrated approach to monitor nutrient dynamics in their own production systems to assess potential nutrient disorders and take timely corrective actions to minimize losses in yield and quality.

## Figures and Tables

**Figure 1 plants-12-00422-f001:**
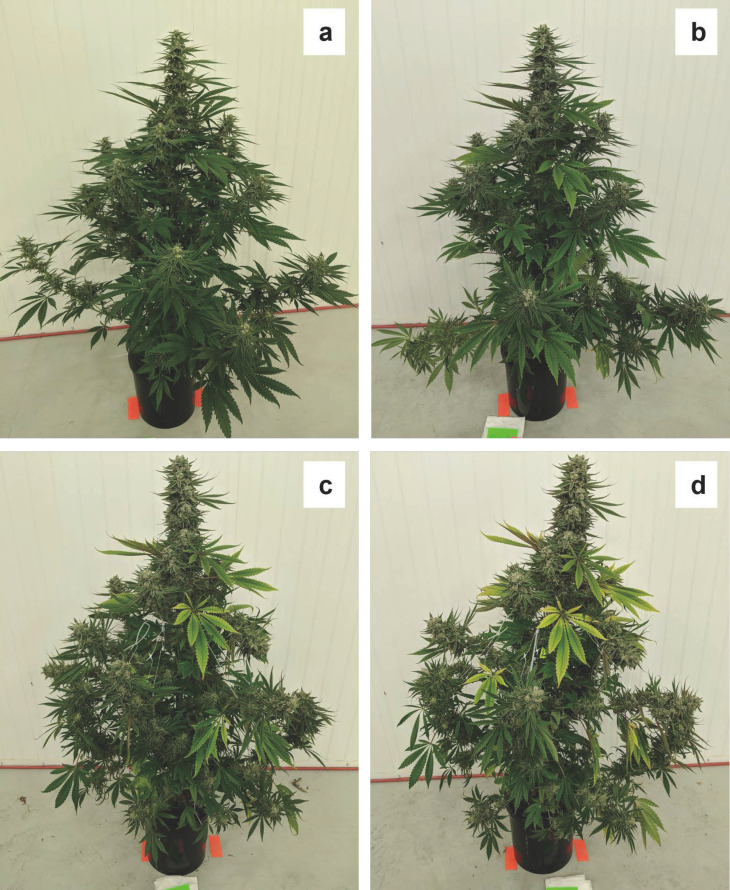
Control plant during (**a**) week five when no foliar deficiency symptoms were visible and (**b**) week six, (**c**) week seven, and (**d**) week eight, as fan leaves developed increasing levels of foliar chlorosis.

**Figure 2 plants-12-00422-f002:**
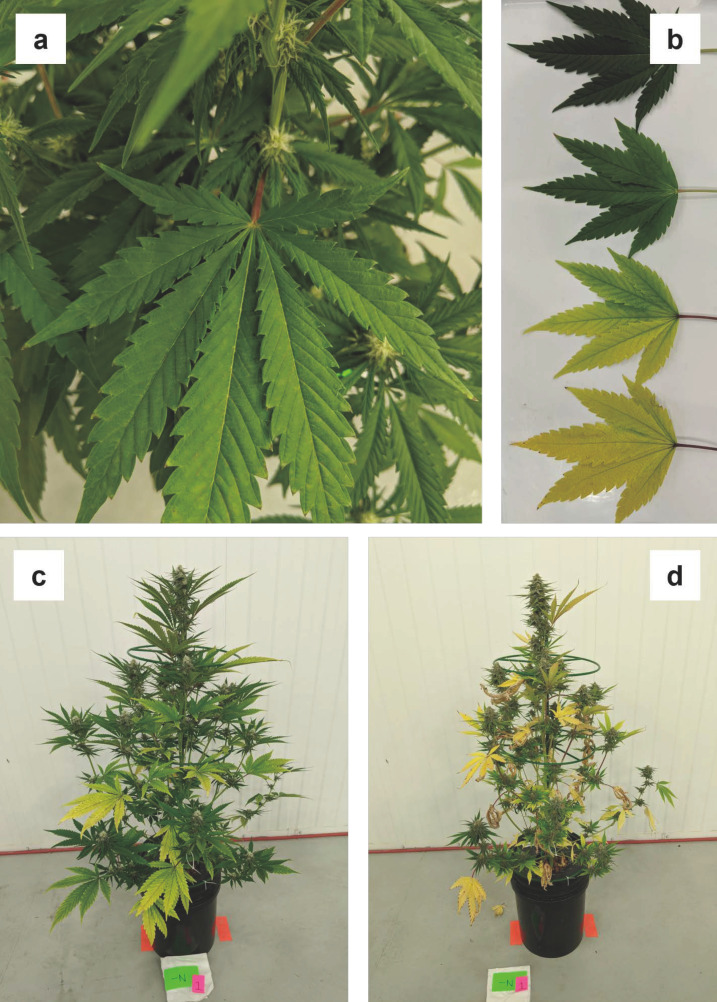
(**a**) Lower canopy foliar tissue of nitrogen (N) deficient plants starting to yellow during week three, (**b**) progression of nitrogen deficiency symptoms on fan leaves from mild (**top**) to severe (**bottom**) during week four, (**c**) whole leaf chlorosis developing in fan leaves of the lower two-thirds of the canopy by week four, and (**d**) almost entire senescence of fan leaves, stem browning and yellowing, and beginning of sugar leaf chlorosis in nitrogen deficient plants during week five.

**Figure 3 plants-12-00422-f003:**
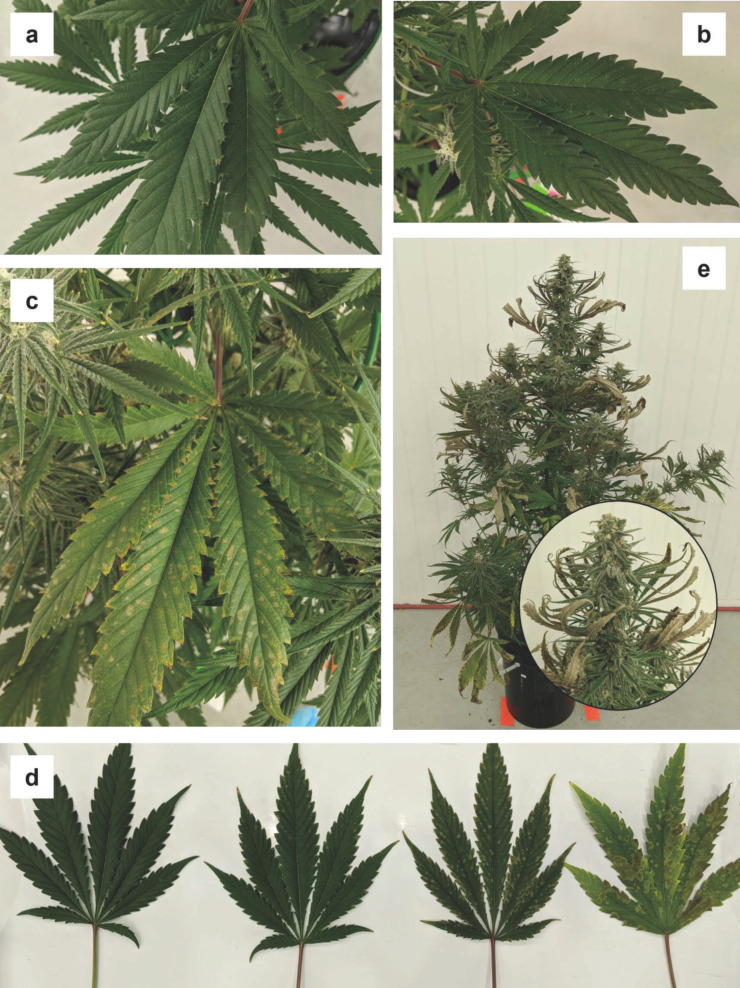
Progression of foliar phosphorous (P) deficiency symptoms in fan leaves during week three (**a**), week four (**b**), and week five (**c**,**d**) and phosphorous deficiency symptoms on the whole plant (**e**) and sugar leaves associated with the apical inflorescence ((**e**), inset).

**Figure 4 plants-12-00422-f004:**
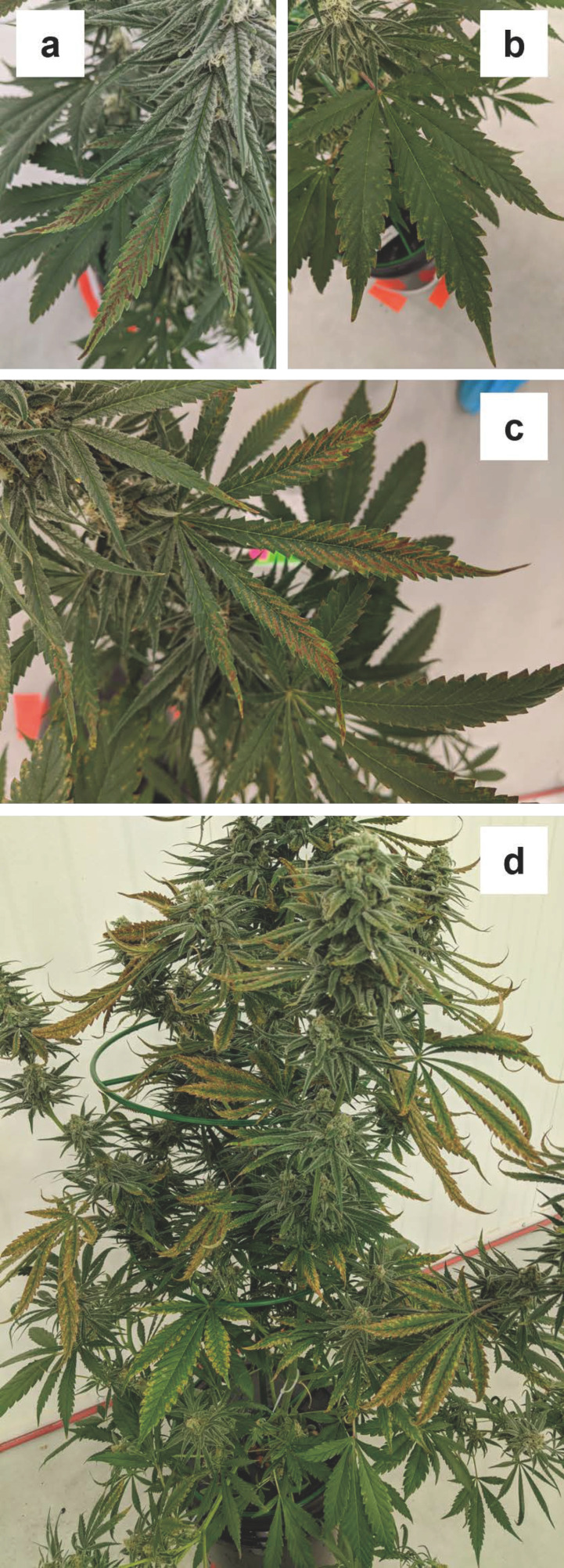
Progression of potassium (K) deficiency on (**a**) sugar, and (**b**) fan leaves on week four and in the middle of the canopy in week five (**c**,**d**).

**Figure 5 plants-12-00422-f005:**
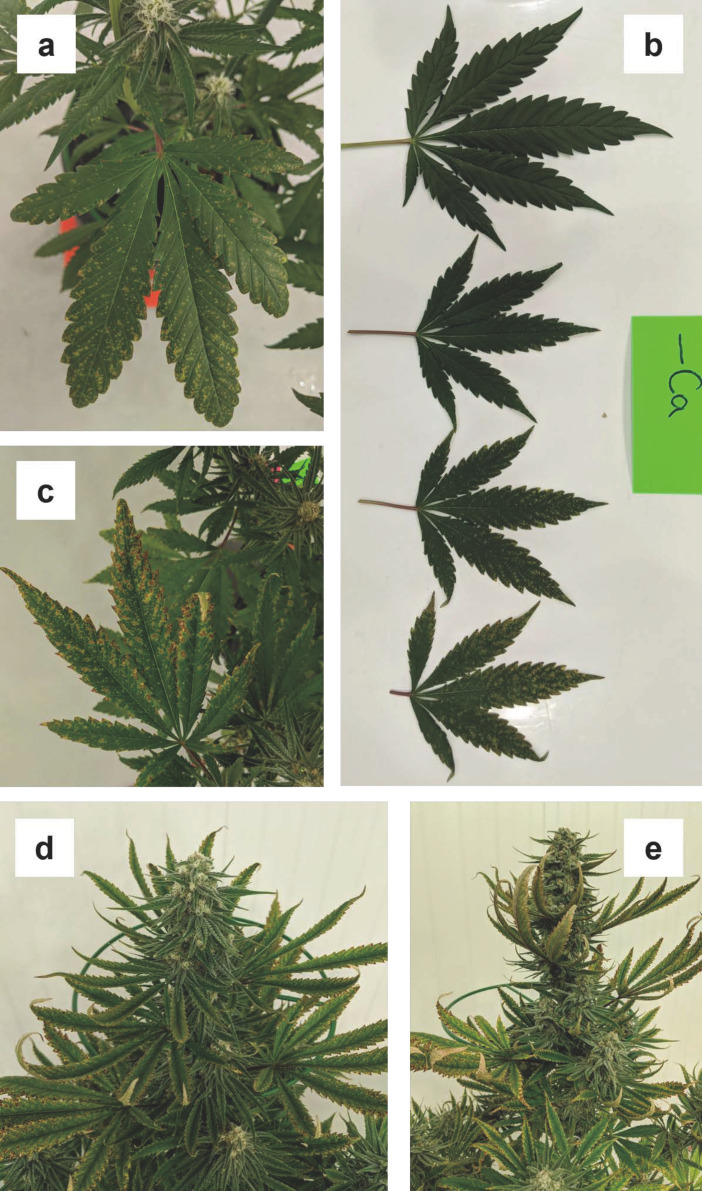
Progression of calcium (Ca) deficiency, (**a**) starting on lower fan leaves in week three, with severity of symptoms being concentrated near leaf margins (**b**). By week five there was general leaf margin necrosis on lower fan leaves (**c**) an upper fan and sugar leaves started to show symptoms (**d**). By week six, the majority of fan and sugar leaves showed characteristic leaf margin necrosis and upper leaves were curled upwards (**e**).

**Figure 6 plants-12-00422-f006:**
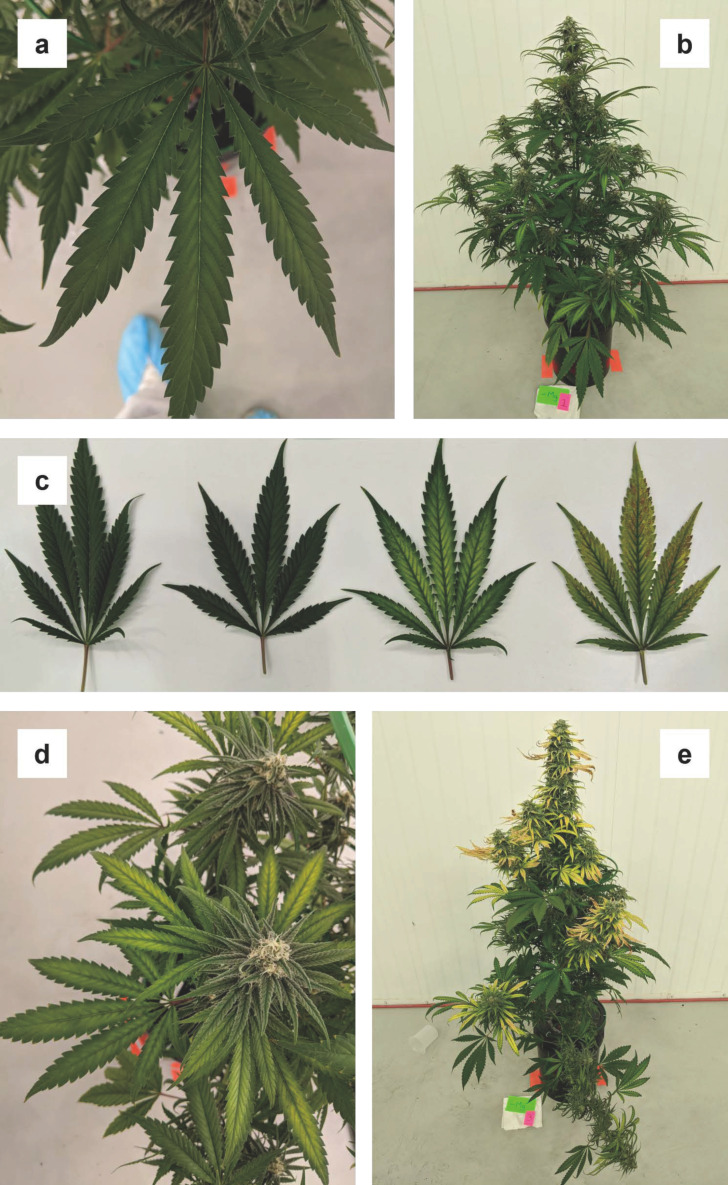
Progression of magnesium deficiency, first appearing in week four (**a**), with interveinal chlorotic (**b**) moving towards necrotic areas in between the leaflet midrib and margins (**c**). In week five, most of the smaller fan leaves started to show symptoms (**d**), and by week six, magnesium deficiency was widespread across the entire plant, except for some of the older and larger fan leaves (**e**).

**Figure 7 plants-12-00422-f007:**
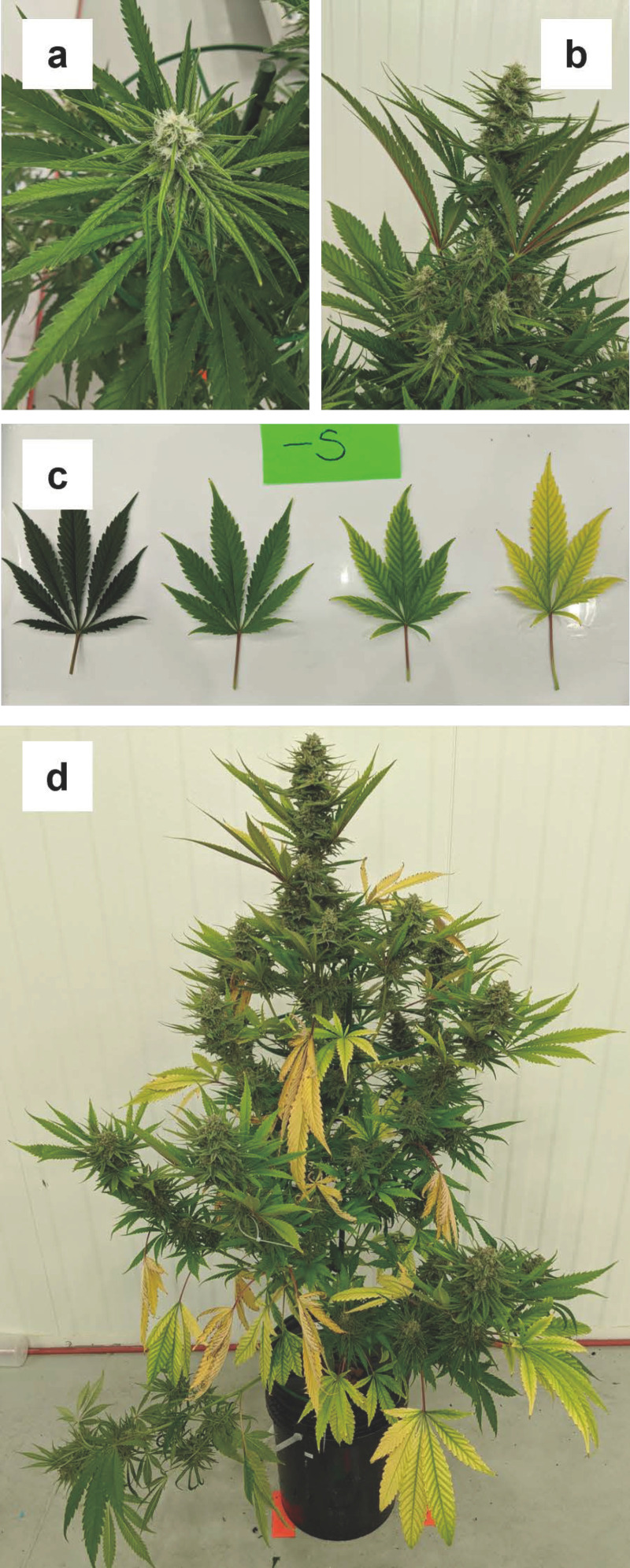
Progression of sulfur deficiency which first appeared as yellowing on the younger sugar leaves (**a**) and upward angled upper canopy fan leaves (**b**). In the following weeks, sulfur deficiency symptoms did not progress in the sugar leaves rather they progressed in fan leaves (**c**), as interveinal chlorosis, and were most severe in lower canopy fan leaves (**d**).

**Figure 8 plants-12-00422-f008:**
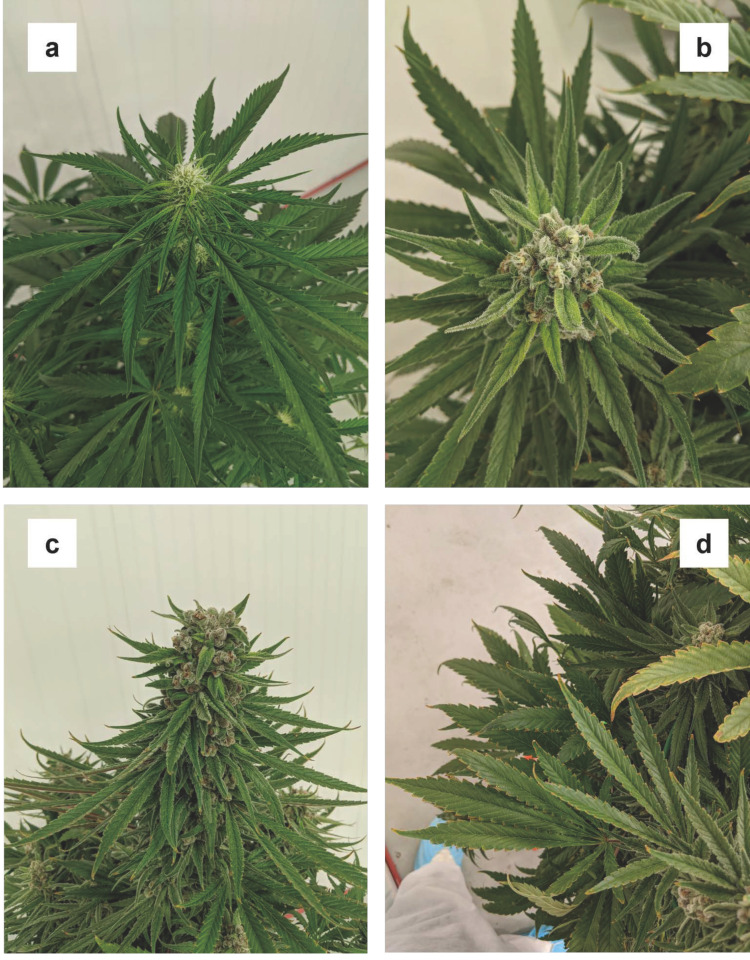
Progression of the various disparate symptoms iron deficiency on upper and lower canopy tissues throughout the trial: mild chlorosis on recently developed sugar leaves (**a**), mottling and chlorosis in the central areas of sugar leaves in week five (**b**), which disappeared in latter weeks (**c**), and edge necrosis on lower fan leaves in week six (**d**).

**Figure 9 plants-12-00422-f009:**
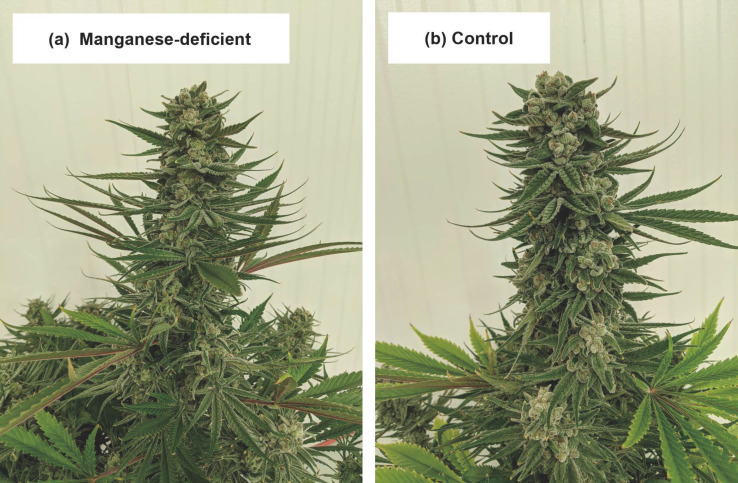
Comparison of apical inflorescence tissues between plants in the manganese deficiency treatment (**a**) and control treatment (**b**) during week seven of the trial.

**Table 1 plants-12-00422-t001:** The calculated solution concentrations (mg∙L^−1^) of nutrient elements (plus Na) in the vegetative-stage nutrient solution and each of the nutrient deficiency treatments. Shaded cells highlight the elemental concentrations of the deficient nutrient in the respective treatments.

Element	Estimated Solution Concentrations (mg∙L^−1^)									
	Veg ^z^	Control	-N ^y^	-P	-K	-Ca	-Mg	-S	-Fe	-Mn
N	130	93	0	93	93	93	93	93	93	93
P	40	36	36	0	36	36	36	36	36	36
K	180	140	98	140	0	140	140	140	140	140
Ca	130	130	67	130	130	0	130	130	130	130
Mg	44	40	24	40	40	40	0	40	40	40
S	59	54	32	54	54	54	54	0	54	54
Fe	2.1	2.3	2.3	2.3	2.3	2.3	2.3	2.3	0	2.3
Mn	0.6	0.73	0.73	0.73	0.73	0.73	0.73	0.73	0.73	0
Zn	0.12	0.11	0.11	0.11	0.11	0.11	0.11	0.11	0.11	0.11
Cu	0.03	0.04	0.04	0.04	0.04	0.04	0.04	0.04	0.04	0.04
B	0.39	0.42	0.42	0.42	0.42	0.42	0.42	0.42	0.42	0.42
Mo	0.02	0.03	0.03	0.03	0.03	0.03	0.03	0.03	0.03	0.03
Cl	28	0.52	130	12	0.52	0.52	0.52	58	0.52	0.06
Na	0	19	19	0.01	100	96	58	19	19	19

^z^ nutrient solution used during the 16 d post-transplant vegetative stage (18 h photoperiod). ^y^ the “-” sign denotes the missing element in each of the respective treatments.

**Table 2 plants-12-00422-t002:** Concentrations of single salt stock solutions and the composition of each of the treatment solutions.

Fertilizer Salt	Formula	Stock Solution Molarity (mol∙L^−1^)	Volume of Stock Solution Added (mL/60 L)								
			Control	-N ^z^	-P	-K	-Ca	-Mg	-S	-Fe	-Mn
Potassium nitrate	KNO_3_	1	200		200		200	200	200	200	200
Calcium nitrate tetrahydrate	Ca(NO_3_)_2_∙ 4H_2_O	2	100		100	100		100	100	100	100
Potassium phosphate monobasic	KH_2_PO_4_	1	20	20			20	20	20	20	20
Magnesium sulfate heptahydrate	MgSO_4_∙7H_2_O	1	100	60	100	100	100			100	100
Potassium chloride	KCl	1		130	20						
Calcium chloride dihydrate	CaCl_2_∙2H_2_O	1		100							
Sodium nitrate	NaNO_3_	1				200	200				
Sodium phosphate monobasic monohydrate	NaH_2_PO_4_∙H_2_O	1	50	50	0	70	50	50	50	50	50
Sodium sulfate anhydrous	Na_2_SO_4_	1						100			
Magnesium chloride hexahydrate	MgCl_2_∙6H_2_O	1							100		
Iron diethylenetriaminepentaacetic acid	FeEDTA	0.1	25	25	25	25	25	25	25		25
Manganese chloride tetrahydrate	MnCl_2_∙4H_2_O	0.04	20	20	20	20	20	20	20	20	
Zinc chloride	ZnCl_2_	0.02	5	5	5	5	5	5	5	5	5
Copper sulfate	CuSO_4_	0.02	2	2	2	2	2	2	2	2	2
Boric acid	H_3_BO_3_	0.1	25	25	25	25	25	25	25	25	25
Sodium molybdate dihydrate	Na_2_MoO_4_∙ 2H_2_O	0.01	2	2	2	2	2	2	2	2	2

^z^ the “-” sign denotes the missing element in each of the respective treatments.

**Table 3 plants-12-00422-t003:** Concentrations (mg∙L^−1^) of nutrient elements (plus Na) and electrical conductivity (EC, µS∙cm^−1^) in the rainwater and each of the nutrient treatments according to third party laboratory analysis of freshly made batches of solutions. Shaded cells highlight the elemental concentrations of the deficient nutrient in the respective treatments.

Element	Measured Solution Concentrations (mg∙L^−1^)									
	Rain-Water	Control	-N ^z^	-P	-K	-Ca	-Mg	-S	-Fe	-Mn
N	7	130	4	130	120	130	120	120	120	130
P	<1	35	34	<1	34	32	35	31	34	32
K	<1	130	87	140	5	130	130	130	130	130
Ca	9.1	130	61	140	130	9.7	120	120	120	120
Mg	0	39	21	39	40	35	1.8	35	39	37
S	6.4	54	58	54	52	50	49	2.9	53	50
Fe	0.19	0.93	1.3	1.2	0.77	1.6	0.86	1.6	0.01	1.4
Mn	0.04	0.76	0.76	0.78	0.76	0.74	0.74	0.65	0.73	0.02
Zn	0.38	0.43	0.41	0.44	0.43	0.26	0.45	0.14	0.43	0.41
Cu	<0.01	0.03	0.04	0.03	0.03	0.02	0.04	0.04	0.03	0.04
B	<0.01	0.43	0.43	0.43	0.43	0.41	0.43	0.39	0.44	0.41
Mo	<0.01	0.02	0.03	0.02	0.03	0.02	0.02	0.02	0.03	0.01
Na	3.1	21	20	4	96	160	90	18	20	19
Cl	17	10	130	23	11	12	11	110	10	9
EC (dS∙m^−1^)	0.1	1.6	0.9	1.6	1.4	1.6	1.5	1.6	1.5	1.6

^z^ the “-” sign denotes the missing element in each of the respective treatments.

**Table 4 plants-12-00422-t004:** Elemental concentrations (% of DW or μg∙g^−1^) in foliar tissues from the upper and lower canopy of each of the nutrient treatments, sampled during the week that deficiency symptoms in each respective plant appeared.

Element Concentration	Location on Plant	Treatment								
		Control	-N ^z^	-P	-K	-Ca	-Mg	-S	-Fe	-Mn
N (%)	lower	2.2 ± 0.14 ^y^ a ^x^	1.9 ± 0.06 a	2.6 ± 0.02 a	3.9 ± 0.20 a	3.2 ± 0.14 a	3.7 ± 0.24 a	3.6 ± 0.42 a	2.3 ± 0.12 a	2.6 ± 0.25 a
	upper	2.6 ± 0.14 a	2.5 ± 0.25 a	2.5 ± 0.17 a	3.7 ± 0.86 a	3.4 ± 0.37 a	3.5 ± 0.31 a	1.6 ± 0.14 a	2.3 ± 0.11 a	2.5 ± 0.17 a
P (%)	lower	0.59 ± 0.006 b	0.41 ± 0.022 a	0.087 ± 0.0018 a	0.78 ± 0.163 a	0.67 ± 0.017 b	0.62 ± 0.067 a	0.86 ± 0.037 a	0.54 ± 0.065 a	0.54 ± 0.036 a
	upper	0.73 ± 0.021 a	0.51 ± 0.055 a	0.13 ± 0.017 a	0.69 ± 0.233 a	0.93 ± 0.062 a	0.58 ± 0.097 a	0.58 ± 0.018 b	0.72 ± 0.089 a	0.67 ± 0.038 a
K (%)	lower	2.2 ± 0.08 a	2.1 ± 0.15 a	2.5 ± 0.14 a	0.84 ± 0.169 a	3.9 ± 0.41 a	2.6 ± 0.30 a	2.4 ± 0.53 a	3.1 ± 0.07 a	3.1 ± 0.19 a
	upper	2.0 ± 0.12 a	1.9 ± 0.11 a	2.7 ± 0.33 a	1.2 ± 0.17 a	3.2 ± 0.20 a	3.1 ± 0.36 a	1.6 ± 0.27 a	2.4 ± 0.22 b	1.6 ± 0.05 a
Ca (%)	lower	7.7 ± 0.40 a	2.3 ± 0.31 a	5.4 ± 0.39 a	4.2 ± 0.84 a	2.7 ± 0.34 a	6.4 ± 0.15 a	4.0 ± 0.31 b	8.2 ± 0.63 a	8.9 ± 0.49 a
	upper	6.3 ± 0.30 b	1.3 ± 0.14 b	3.9 ± 0.61 a	4.7 ± 0.29 a	0.97 ± 0.159 b	7.8 ± 1.21 a	6.4 ± 0.59 a	6.7 ± 0.50 a	7.6 ± 0.67 a
Mg (%)	lower	1.2 ± 0.02 a	0.40 ± 0.026 a	0.92 ± 0.082 a	0.75 ± 0.174 a	1.6 ± 0.12 a	0.097 ± 0.0127 b	0.61 ± 0.068 b	1.2 ± 0.15 a	1.2 ± 0.05 a
	upper	0.98 ± 0.012 b	0.32 ± 0.012 a	0.63 ± 0.082 a	0.75 ± 0.069 a	1.9 ± 0.46 a	0.32 ± 0.063 a	0.87 ± 0.059 a	1.2 ± 0.12 a	1.2 ± 0.07 a
S (%)	lower	0.34 ± 0.009 a	0.27 ± 0.040 a	0.25 ± 0.008 a	0.26 ± 0.042 a	0.33 ± 0.014 a	0.28 ± 0.050 a	0.14 ± 0.022 a	0.36 ± 0.03 a	0.32 ± 0.021 a
	upper	0.34 ± 0.011 a	0.27 ± 0.020 a	0.25 ± 0.041 a	0.26 ± 0.053 a	0.31 ± 0.035 a	0.31 ± 0.079 a	0.083 ± 0.0132 a	0.51 ± 0.04 b	0.32 ± 0.025 a
Fe (μg∙g^−1^)	lower	130 ± 9.1 a	140 ± 15 a	310 ± 44 a	110 ± 23 a	71 ± 14.5 a	50 ± 3.1 a	100 ± 9.2 a	64 ± 10.1 a	150 ± 24 a
	upper	99 ± 7.4 b	170 ± 9 a	230 ± 58 a	78 ± 18.6 a	54 ± 7.0 a	65 ± 10.5 a	120 ± 15 a	36 ± 2.5 a	93 ± 14.2 a
Mn (μg∙g^−1^)	lower	330 ± 26 a	170 ± 7 a	330 ± 8 a	250 ± 75 a	320 ± 55 a	230 ± 7 a	140 ± 13 a	390 ± 60 a	160 ± 11 a
	upper	210 ± 40 a	140 ± 10 a	260 ± 53 a	260 ± 48 a	270 ± 63 a	360 ± 60 a	200 ± 23 a	360 ± 47 a	10 ± 1.0 b
Zn (μg∙g^−1^)	lower	100 ± 2 a	74 ± 11.7 a	110 ± 7 a	83 ± 21.1 a	70 ± 16.3 a	62 ± 4.7 a	43 ± 2.7 a	110 ± 17 a	61 ± 2.4 b
	upper	100 ± 4 a	95 ± 6.6 a	78 ± 1.5 a	78 ± 22.8 a	97 ± 25.2 a	58 ± 13.0 a	29 ± 2.4 b	160 ± 21 a	74 ± 3.6 a
Cu (μg∙g^−1^)	lower	4.2 ± 0.37 a	4.2 ± 0.21 b	5.7 ± 0.46 a	7.9 ± 1.29 a	6.2 ± 0.74 a	7.5 ± 0.56 a	8.3 ± 1.07 a	4.3 ± 0.17 a	4.3 ± 0.70 a
	upper	5.3 ± 0.39 a	7.2 ± 0.24 a	6.5 ± 0.30 a	7.6 ± 1.89 a	7.2 ± 1.62 a	6.9 ± 0.62 a	3.7 ± 0.51 b	6.4 ± 1.00 a	5.0 ± 0.29 a
B (μg∙g^−1^)	lower	190 ± 6.7 a	89 ± 6.2 a	230 ± 16 a	140 ± 19 a	140 ± 11 a	180 ± 11 a	130 ± 13 a	170 ± 31 a	190 ± 15 a
	upper	200 ± 8.8 a	93 ± 3.2 a	170 ± 5 b	140 ± 24 a	150 ± 27 a	180 ± 25 a	130 ± 9 a	200 ± 18 a	240 ± 19 a
Al (μg∙g^−1^)	lower	47 ± 8.7 a	14 ± 1.0 a	14 ± 2.1 a	19 ± 3.7 a	15 ± 4.6 a	23 ± 2.3 a	19 ± 6.4 a	47 ± 5.9 a	39 ± 2.8 a
	upper	27 ± 1.8 a	16 ± 4.2 a	11 ± 0.8 a	18 ± 0.7 a	18 ± 4.5 a	27 ± 2.0 a	38 ± 3.7 a	40 ± 0.9 a	32 ± 1.1 a
Na (μg∙g^−1^)	lower	0.012 ± 0.0006 a	0.013 ± 0.0012 a	0.013 ± 0.0012 a	0.011 ± 0.0000 a	0.021 ± 0.0015 a	0.011 ± 0.0007 a	0.010 ± 0.0006 a	0.010 ± 0.0033 a	0.012 ± 0.000 a
	upper	0.013 ± 0.0012 a	0.014 ± 0.0012 a	0.013 ± 0.0017 a	0.012 ± 0.0026 a	0.012 ± 0.0012 a	0.011 ± 0.0007	0.011 ± 0.0009 a	0.0097 ± 0.00033 a	0.013 ± 0.001 a

^z^ the “-” sign denotes the missing element in each of the respective treatments. ^y^ data are means ± SE (*n* = 3). ^x^ within each element in each treatment, means followed by the same letter indicate no differences between upper- and lower-canopy concentrations at *p* ≤ 0.05 according to Tukey’s honesty significant difference test.

**Table 5 plants-12-00422-t005:** Fresh weight (FW) of separated aboveground tissues, harvest index (inflorescence FW/stem and leaves FW), and the dry weight (DW) of the roots from each nutrient treatment.

Treatment	Stem and Leaves FW (g)	Inflorescence FW (g)	Harvest Index	Root DW (g)
Control	567 ± 23.6 ^z^ ab ^y^	1110 ± 68.8 a	0.66 ± 0.01 a	26.2 ± 1.67 abc
-N ^x^	155 ± 21.4 c	315 ± 17.4 e	0.67 ± 0.02 a	13.2 ± 1.52 d
-P	229 ± 21.2 bc	406 ± 26.1 de	0.64 ± 0.02 a	36.2 ± 2.32 a
-K	349 ± 48.4 abc	535 ± 27.1 cde	0.61 ± 0.03 a	16.8 ± 1.85 cd
-Ca	474 ± 64.2 abc	621 ± 74.5 cde	0.57 ± 0.04 a	25.5 ± 2.49 abc
-Mg	486 ± 163 abc	748 ± 53.4 bc	0.62 ± 0.09 a	24.0 ± 4.00 bcd
-S	485 ± 104 abc	731 ± 12.4 cd	0.61 ± 0.05 a	25.3 ± 1.19 abc
-Fe	621 ± 58.5 a	1060 ± 61.3 ab	0.63 ± 0.02 a	31.6 ± 3.42 ab
-Mn	602 ± 36.5 a	1060 ± 145 ab	0.63 ± 0.02 a	27.7 ± 1.89 abc

^z^ data are means ± SE (*n* = 3). ^y^ means, within each column, followed by the same letter are not different at *p* ≤ 0.05 according to Tukey’s honesty significant difference test. ^x^ the “-” sign denotes the missing element in each of the respective treatments.

**Table 6 plants-12-00422-t006:** Concentrations (mg∙g^−1^ of dry tissue) of cannabinoids measured in apical inflorescence tissues of each of the nutrient deficiency treatments.

Cannabinoid ^z^	Treatment								
	Control	-N ^y^	-P	-K	-Ca	-Mg	-S	-Fe	-Mn
CBC	0.45 ± 0.006 ^x^ ab ^w^	0.45 ± 0.049 ab	0.53 ± 0.012 a	0.46 ± 0.011 ab	0.32 ± 0.038 b	0.45 ± 0.060 ab	0.37 ± 0.062 ab	0.35 ± 0.041 ab	0.42 ± 0.003 ab
CBD	0.39 ± 0.007 a	0.39 ± 0.053 a	0.25 ± 0.124 a	0.43 ± 0.006 a	0.39 ± 0.027 a	0.44 ± 0.038 a	0.40 ± 0.037 a	0.40 ± 0.048 a	0.33 ± 0.003 a
CBDA	0.35 ± 0.012 ab	0.34 ± 0.018 ab	0.36 ± 0.029 ab	0.39 ± 0.013 a	0.23 ± 0.042 b	0.30 ± 0.032 ab	0.31 ± 0.053 ab	0.28 ± 0.017 ab	0.31 ± 0.006 ab
T-CBD	0.68 ± 0.018 a	0.70 ± 0.037 a	0.56 ± 0.148 a	0.77 ± 0.009 a	0.59 ± 0.015 a	0.70 ± 0.042 a	0.67 ± 0.018 a	0.65 ± 0.035 a	0.60 ± 0.010 a
CBG	0.98 ± 0.012 a	0.84 ± 0.072 ab	0.85 ± 0.033 ab	0.82 ± 0.023 ab	0.61 ± 0.064 b	0.77 ± 0.059 ab	0.86 ± 0.091 ab	0.80 ± 0.106 ab	0.90 ± 0.023 ab
CBGA	3.7 ± 0.28 abc	3.0 ± 0.07 abcd	3.1 ± 0.18 abcd	4.0 ± 0.26 a	2.2 ± 0.19 d	2.8 ± 0.07 bcd	2.5 ± 0.14 cd	3.2 ± 0.52 abcd	3.8 ± 0.08 ab
CBN	nd ^v^	0.17 ± 0.091 a	0.29 ± 0.007 a	0.11 ± 0.107 a	nd	0.20 ± 0.102 a	0.20 ± 0.098 a	nd	0.18 ± 0.090 a
d8THC	0.70 ± 0.010 ab	0.83 ± 0.047 ab	0.66 ± 0.023 b	0.75 ± 0.012 ab	0.73 ± 0.036 ab	0.72 ± 0.061 ab	0.88 ± 0.006 a	0.71 ± 0.073 ab	0.71 ± 0.038 ab
THC	7.3 ± 0.31 ab	7.9 ± 1.31 ab	9.2 ± 0.07 ab	6.2 ± 0.50 ab	4.3 ± 1.08 b	11 ± 1.5 a	6.1 ± 1.62 ab	5.5 ± 0.97 b	7.2 ± 0.36 ab
THCA	183 ± 3.6 ab	177 ± 6.2 ab	180 ± 11.1 ab	204 ± 6.4 a	134 ± 16.2 b	159 ± 12.9 ab	166 ± 20.4 ab	158 ± 7.7 ab	171 ± 4.3 ab
T-THC	168 ± 3.22 ab	163 ± 6.7 ab	167 ± 9.8 ab	185 ± 5.7 a	122 ± 15.3 b	150 ± 12.8 ab	151 ± 19.5 ab	144 ± 7.7 ab	157 ± 4.1 ab

^z^ CBC, cannabichromene; CBD, cannabidiol; CBDA, cannabidiolic acid; T-CBD, total equivalent CBD; CBG, cannabigerol; CBGA, cannabigerolic acid; CBN, cannabinol; d8THC, Δ^8^-tetrahydrocannabinol; THC, Δ^9^-tetrahydrocannabinol; THCA, Δ^9^-tetrahydrocannabinolic acid; T-THC, total equivalent Δ^9^-tetrahydrocannabinol. ^y^ the “-” sign denotes the missing element in each of the respective treatments. ^x^ data are means ± SE (n = 3). ^w^ means, within each row, followed by the same letter are not different at *p* ≤ 0.05 according to Tukey’s honesty significant difference test. ^v^ not detected.

## Data Availability

All data from the study are included in the manuscript, further inquiries can be directed to the corresponding author.

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
