# Peer review of "Foliar Symptomology, Nutrient Content, Yield, and Secondary Metabolite Variability of Cannabis Grown Hydroponically with Different Single-Element Nutrient Deficiencies"

_plants, 2023, doi:10.3390/plants12030422_

Round 1

Reviewer 1 Report

The plants-2101397 with the title of Foliar symptomology, nutrient content, yield, and secondary metabolite variability of cannabis grown hydroponically with different single-element nutrient deficiencies  investigate an interesting topic, but the authors should revise the following comments to make it suitable for publication in Plants.

Please follow the guidelines of authors and format of the journal, particularly citations and other issues.

Abstract: please add the most important finding in terms of some values.

Introduction:

L 40: and many other factors? What are they? We as readers do not know what do you mean? Please mention them.

Please add one-two sentences in the introduction about the ability of cannabis to accumulate heavy metals using these ref: https://doi.org/10.1016/j.chemosphere.2012.07.031

L46-50 cite this text please

L54-57 the authors said>> Prior studies have illustrated some>> and at the end of the sentences, there is only one citation>> Please add at least two citations for this sentence.

L66-85 cite this text please, you only cited one ref for this paragraph. This is not fair.

Sometimes, I feel that there is jumping in the text within the introduction section. Please kindly reorganize this section.

Material and methods:

Well written, but the authors should cite all used methods in this sections since those methods were not their own.

Results and Discussion:

I prefer authors divide this section into two sections as following

3. Results

4. Discussion

The discussion of this ms is not strong enough, the authors should focus on the mechanisms of their treatments using why and how?

I am wondering why authors presented their elemental analysis as μg∙g-1? example Table 4! Why not were presented in g∙kg-1 or mg∙kg-1 depending on the element?

If there were not significant differences among different treatments for some measurements, then why the authors present the same letters after the means of treatments? What does this show?

What do think, why there were no significant differences among your treatments? Can you explain this in details within your ms.

Author Response

We have revised the manuscript according to the reviewers’ suggestions using “track Changes” and have responded to the reviewers’ comments below, point-by-point, below. We have also reformatted the citation style from ‘name-date’ to’ order of first use’, according to the Assistant Editor’s (Ms. Arielle Foo) suggestions.

Reviewer 1

The plants-2101397 with the title of Foliar symptomology, nutrient content, yield, and secondary metabolite variability of cannabis grown hydroponically with different single-element nutrient deficiencies  investigate an interesting topic, but the authors should revise the following comments to make it suitable for publication in Plants.

Thank you for taking the time and effort to review our manuscript and help us make it better. We have made revisions according to your recommendations, which we have outlined point-by-point below.

Please follow the guidelines of authors and format of the journal, particularly citations and other issues.

Yes, we have reformatted the manuscript based on your suggestion and by consulting with the journal (via Ms Foo). The citations have been converted to numerical order of appearance in the revised manuscript. We have been advised that the current order of manuscript subsections (e.g., Materials and Methods located prior to Results and Discussion) is acceptable by Plants.

Abstract: please add the most important finding in terms of some values.

Following your suggestions, we have added some values showing the magnitude of reduction of vegetative and floral biomass in the deficiency treatments relative to the control. In the context of our objectives, the most important findings are the descriptions and images of the onset and progression of deficiency symptomology – which are difficult to summarize and can’t be enumerated.

Introduction:

L 40: and many other factors? What are they? We as readers do not know what do you mean? Please mention them.

The phrase “, and many other factors” was ambiguous, so it has been deleted.

Please add one-two sentences in the introduction about the ability of cannabis to accumulate heavy metals using these ref: https://doi.org/10.1016/j.chemosphere.2012.07.031

We added a section in the introduction regarding the effects of individual elements, including heavy metals, on the quality and marketability of cannabis tissues. We cited the mentioned reference.

L46-50 cite this text please

We added a citation

L54-57 the authors said>> Prior studies have illustrated some>> and at the end of the sentences, there is only one citation>> Please add at least two citations for this sentence.

Since this was an introductory statement, no citations were added for the first sentence. However, the appropriate citations are provided at the end of their respective sentences throughout the rest of the paragraph. We believe it would be unwieldy to also provide all of these citations at the end of the first sentence.

L66-85 cite this text please, you only cited one ref for this paragraph. This is not fair.

We had intentionally chosen to not cite all relevant publications in this paragraph, because it could  generally be perceived as being overly critical of these prior works, particularly since the majority of them came from a single research group. This didn’t seem fair, since the objectives of these studies were not necessarily to identify and describe cannabis foliar nutrient deficiencies. Therefore, to not single out specific prior publications (all of which were already cited elsewhere) for their limitations, we opted to generalize and had worded the text accordingly. However, we have added these citations in the revised manuscript.

Sometimes, I feel that there is jumping in the text within the introduction section. Please kindly reorganize this section.

We have rearranged some of the text in the introduction to improve flow and clarity.

Material and methods:

Well written, but the authors should cite all used methods in this sections since those methods were not their own.

To the extent possible within the limitations of the independent labs’ disclosure protocols for proprietary information, we have provided additional information regarding the analytical methods used for analysis of nutrient solutions and cannabis tissues. The specific details on analytical methods used in cannabis are proprietary in Canada, however independent labs’ methodologies must adhere to strict guidelines and are validated and approved by federal regulating bodies.

Results and Discussion:

I prefer authors divide this section into two sections as following

  1. Results
  2. Discussion

At the manuscript drafting stage, we had carefully considered separately reporting the Results and Discussion but decided against doing so. While this is somewhat of a stylistic decision, we felt that combining these two sections - which is permitted by Plants - improved readability, flow and comprehension.

The discussion of this ms is not strong enough, the authors should focus on the mechanisms of their treatments using why and how?

Mechanistic investigations of cannabis’ responses to single-element nutrient deficiencies was not among the objectives of the study reported in this manuscript. Conducting such an investigation would have been well beyond the logistical and operational constraints of this study as it would have required a much larger growing area, more plant material, and considerably more rigorous evaluations of the spatial and temporal dynamics of both individual nutrient elements and secondary metabolites in all different cannabis tissues (i.e., root, stem, foliar, and floral tissues). While these would be excellent topics for future study, we will not further conjecture, based on the results of this study, about the potential mechanisms of cannabis’ responses to single-element nutrient deficiencies.   

I am wondering why authors presented their elemental analysis as μg∙g-1? example Table 4! Why not were presented in g∙kg-1 or mg∙kg-1 depending on the element?

We found no standard unit conventions for cannabis tissue nutrient (or secondary metabolite) composition in the studies we cited, therefore we reported the foliar tissue element concentrations in the units that were provided in the analysis reports from the independent laboratory. Since these units are clearly defined in the table caption, readers can easily convert to whatever unit convention best suits their needs.

If there were not significant differences among different treatments for some measurements, then why the authors present the same letters after the means of treatments? What does this show?

The tables’ footnotes clearly describe the meaning of what the same letter following different means shows.  For example, in Table 4: “xwithin each element in each treatment, means followed by the same letter indicate no differences between upper- and lower-canopy concentrations at P ≤ 0.05 according to Tukey’s honesty significant difference test.”

For consistency, we have chosen to maintain this convention throughout all tables where multiple comparisons data was presented, regardless of whether there were treatment effects in individual statistical comparisons

What do think, why there were no significant differences among your treatments? Can you explain this in details within your ms.

Since we present and discuss many significant nutrient deficiency treatment effects and there are naturally no treatment effects in every statistical comparison that was done. To avoid too much speculations, we are more focusing on discussion on the main objectives of this study.

Thanks again for your useful and helpful suggestions!

Reviewer 2 Report

Thank you for entrusting me with the review of this interesting work. The work is written in an appropriate language and its layout is consistent with the framework of the journal. The article is innovative, however, it deals with very topical issues. The work contains many interesting results in clear tables and many photos that increase its value. However, before publication, please add a few items of literature in the discussion or introduction chapter.

Author Response

Thank you for reviewing our manuscript, we appreciate your objective statements on the organization and quality of the content. We have added some more citations, reorganised some contents and improved the presentation further based on both yours and the other reviewers. 

We have revised the manuscript according to the suggestions by the reviewers and the Assistant Editor’s (Ms. Arielle Foo) using “track Changes”. Please see the attached revised manuscript.

Round 2

Reviewer 1 Report

The ms has been improved